# Contemporary Use of ICSI and Epigenetic Risks to Future Generations

**DOI:** 10.3390/jcm11082135

**Published:** 2022-04-11

**Authors:** Romualdo Sciorio, Sandro C. Esteves

**Affiliations:** 1Edinburgh Assisted Conception Programme, Royal Infirmary of Edinburgh, Edinburgh EH16 4SA, UK; 2Androfert, Andrology and Human Reproduction Clinic, Campinas 13075-460, Brazil; s.esteves@androfert.com.br; 3Department of Surgery, Division of Urology, University of Campinas, Campinas 13083-970, Brazil; 4Faculty of Health, Aarhus University, 8000 Aarhus, Denmark

**Keywords:** assisted reproductive technology, human in vitro fertilization, embryo development, male infertility, intracytoplasmic sperm injection, epigenetics, histone modification, DNA methylation

## Abstract

Since the birth of Louise Brown in 1978 via IVF, reproductive specialists have acquired enormous knowledge and refined several procedures, which are nowadays applied in assisted reproductive technology (ART). One of the most critical steps in this practice is the fertilization process. In the early days of IVF, a remarkable concern was the unpleasant outcomes of failed fertilization, overtaken by introducing intracytoplasmic sperm injection (ICSI), delineating a real breakthrough in modern ART. ICSI became standard practice and was soon used as the most common method to fertilize oocytes. It has been used for severe male factor infertility and non-male factors, such as unexplained infertility or advanced maternal age, without robust scientific evidence. However, applying ICSI blindly is not free of potential detrimental consequences since novel studies report possible health consequences to offspring. DNA methylation and epigenetic alterations in sperm cells of infertile men might help explain some of the adverse effects reported in ICSI studies on reproductive health in future generations. Collected data concerning the health of ICSI children over the past thirty years seems to support the notion that there might be an increased risk of epigenetic disorders, congenital malformations, chromosomal alterations, and subfertility in babies born following ICSI compared to naturally conceived children. However, it is still to be elucidated to what level these data are associated with the cause of infertility or the ICSI technique. This review provides an overview of epigenetic mechanisms and possible imprinting alterations following the use of ART, in particular ICSI. It also highlights the sperm contribution to embryo epigenetic regulation and the risks of in vitro culture conditions on epigenetic dysregulation. Lastly, it summarizes the literature concerning the possible epigenetic disorders in children born after ART.

## 1. Background

Over the past 40 years, assisted reproductive technology (ART) has evolved from an ambitious and experimental procedure to mainstream medicine. This has been obtained thanks to the constant advancements in ovarian stimulation and luteal phase support protocols, sperm preparation techniques, fertilization, and embryo culture methods, and importantly to the progress in cryopreservation of gametes and embryos, which improved pregnancy outcomes and live birth delivery. Worldwide, around 9 million children have been conceived by ART, and more than 3 million cycles are performed globally every year [1,2]. The IVF process is primarily dependent on three procedures: ovarian stimulation (OS), in vitro fertilization (IVF), or ICSI, and embryo culture. However, the process omits critical physiological reproductive steps and it includes a variable degree of invasiveness with unknown consequences. On this basis, the safety of these methods has been questioned. Historically, medically assisted reproduction (MAR) practices have been reported to be safe as most ART babies are healthy [3,4]. However, recent studies report that singletons born following IVF/ICSI treatments have an increased risk of adverse perinatal outcomes, which might be associated with epigenetic dysregulation, such as abnormal placentation or low birth weight [5,6]. Furthermore, there is a raising alertness on the long-term consequences on the health of ART-born children as limited evidence suggests potential adverse metabolic and cardiovascular issues in ART babies compared to naturally conceived children [6,7,8]. Notwithstanding these observations, it is challenging to demonstrate whether the cause of infertility or the fertilization method (i.e., ICSI) plays a role in raising the frequency of specific epigenetic disorders [9]. Most studies are not properly controlled for the influence of paternal age, which is an important factor to consider as with aging sperm methylation could be disrupted and therefore result, for instance, in an increased risk of neurodevelopmental disorders in the offspring [10]. Currently, there is an active debate that ART interventions, such as OS and extended embryo culture to the blastocyst stage, could promote adverse epigenetic effects, considering they occur when most epigenetic reprogramming takes place [11]. Due to the invasiveness of ICSI as a fertilization method, the technique is frequently debated as potentially causing epigenetic dysregulation. The procedure requires the injection of a single sperm cell directly into the oocyte cytoplasm using a narrow and sterile micropipette. This technique introduced in 1992 [12] represents one of the most remarkable changes in the field of MAR, allowing men with low sperm numbers and/or abnormal sperm parameters to become biological fathers [13]. Nowadays, ICSI is an established procedure applied worldwide to treat couples with infertility. However, its application is still the object of debate, particularly concerning its potential adverse consequences on the health of the resulting offspring. In line with that, new data from epigenetics studies highlight possible associations between events occurring in early and adult-onset diseases and male infertility [14]. Male infertility is the primary indication for the treatment in around 30% of couples undergoing ART [15,16]. Given the importance of the sperm epigenome to early embryogenesis, the implications of using sperm from males with fertility problems for ICSI, have to be addressed. Indeed, approximately 15% of male infertility cases involve gene alterations, such as karyotype abnormalities and microdeletions on the Y chromosome, resulting in severe oligozoospermia and azoospermia [17]. The epigenetic regulation of gene activity represents a critical aspect of sperm function and related fertilizing ability [18]. Recent evidence has shown that disruption to the paternal epigenome can induce male infertility and transfer aberrant information to the embryo. One key element of controlling male gamete function involves post-translational modification of histones (PTMs), such as methylation (me), acetylation (ac), and phosphorylation (ph), which allows for activation or repression of underlying genes [19]. Histone PTMs are essential in governing cellular processes, such as transcription, DNA repair, DNA replication, and chromosome condensation [20]. In line with this, a recent study published by Schon and colleagues reported on the overall reduction in *H4* acetylation and alterations in *H4K20* and *H3K9* methylation in asthenoteratozoospermic men compared to normozoospermic men [21]. Moreover, Vieweg and collaborators proposed that abnormal histone acetylation within developmentally important gene promoters in subfertile men could be associated with insufficient sperm chromatin compaction, affecting the transfer of epigenetic marks to the oocyte and the future generation [22,23]. On this basis, ICSI technology, which is largely applied to overcome the most severe forms of male infertility, might increase the frequency of imprinting disorders and adversely affect embryo evolution and future offspring by adopting immature spermatozoa that may not have been adequately imprinted or methylated. This review provides an overview of epigenetic mechanisms and describes possible imprinting alterations following the use of ART, particularly ICSI for male and non-male factor infertility conditions. We also highlight the sperm contribution to embryo epigenetic regulation and the risks of in vitro culture conditions on epigenetic dysregulation. Lastly, we summarize the literature concerning the possible epigenetic disorders in children born after ART, particularly ICSI.

## 2. Overview of Epigenetic Mechanisms

In 1942, Conrad Waddington highlighted the importance of environmentally directed changes during the early stages of mammalian embryo development [24]. Waddington introduced the term “epigenetics”, representing a gene-regulatory mechanism that leads to heritable changes in gene function that are not associated with changes in the DNA sequence. Epigenetic processes include DNA methylation, histone modifications, and chromatin remodeling. These modifications can have short or long-term consequences and be transmitted mitotically from cell to cell and through the germline to the next generation [25,26]. DNA encased around histones results in nucleosome formation, which is part of the chromatin pattern. The particular DNA disposition establishes whether a gene will be transcriptionally active or silent. By contrast, portions of highly compacted DNA are termed heterochromatin and are transcriptionally silent. Tracts that are weakly bonded to histones are called euchromatin and are transcriptionally operative. Epigenetics controls DNA’s compactness and reprogramming, which play a vital role in regulating which genes are active, when they are active, and in what tissues. During gametogenesis and early embryo development, sweeping epigenetic modifications are introduced to both male and female inherited chromatin as two terminally differentiated cells (i.e., the spermatozoon and the oocyte) unite to form a totipotent zygote. Such sweeping modifications to chromatin during this early stage of development render the mammalian genome sensitive to environmentally induced epigenetic changes, which can come in the form of covalent modifications to DNA, associated proteins, and coding and non-coding RNAs [27,28]. Accumulating evidence, mainly from animal studies, indicates that such modifications can, in turn, lead to alterations in developmental processes resulting in congenital abnormalities with a longer-term predisposition to certain diseases in adulthood [29,30,31]. DNA methylation is probably the most explored epigenetic mark [32]. It refers to the addition of a methyl group at the carbon 5 position of the cytosine pyrimidine ring to CG dinucleotide (CpG sites) [33]. Those epigenetic modifications are maintained by daughter cells throughout cell divisions by DNA methyltransferases (DNMTs) [34]. Epigenetic changes are crucial in regulating gene expression during embryo development, whereby any disruption to epigenetic states during this sensitive time window can lead to consequences for development and disease [35,36]. Genomic imprinting is an epigenetic regulation resulting in monoallelic expression of either the maternally or the paternally inherited allele. In humans, this process of parental specific expression is limited to about 200 imprinted genes [37,38,39]. This exclusive mono-allelic expression is under the control of distinctive epigenetic marks and regulatory elements, such as DNA methylation, histone modifications, and long non-coding RNA (lncRNA) [40]. The parental-specific imprints established in the germline escape epigenetic reprogramming in preimplantation embryos, where imprinted genes are important for proper early evolution and are significant for establishing energy equity between the developing fetus and the mother [41]. In humans, genetic alterations, copy number aberrations, and epigenetic modifications affecting imprinted genes have been associated with several diseases, such as Beckwith-Wiedemann syndrome (BWS), Angelman syndrome (AS), Silver-Russell syndrome (SRS), and Prader-Willi syndrome (PWS) [42], characterized by clinical features affecting development, metabolism, and growth (Table 1).

## 3. Imprinting Alteration following ART

After fertilization, the zygote develops into a structure called the “blastocyst”, which includes about 200 cells, already differentiated into two types: the trophectoderm (TE) and the inner cell mass (ICM). The latter comprises a group of cells attached to the inside of the trophectoderm, which will eventually give rise to the fetus. TE cells are the blastocyst’s external layer, promoting the implantation process into the uterine lining and forming other extraembryonic tissues, including the placenta. Embryonic cells are guided toward their future lineages during early development through epigenetic reprogramming and subsequent re-establishment of cell-type-specific epigenetic signatures. This coincides with the period when gametes and embryos are cultivated inside the embryology laboratory. Therefore, during this critical time window, any artificial perturbations might lead to epigenetic modifications in the resultant offspring (Figure 1 and Figure 2). Studies reported imprinted loci to be vulnerable to external environmental cues during in vitro embryo culture. For example, *KvDMR1* has been abnormally methylated in ART-related BWS in humans [42,60,61] and hypomethylated in ART-produced bovine progeny with large offspring syndrome (LOS) [62]. Additionally, several reports indicate that ART-related procedures, including OS, ICSI, and extended culture to the blastocyst stage, might promote epigenetic aberrations [41,42,55]. A review published by Lazaraviciute and colleagues compared the frequency of imprinting alterations and DNA methylation errors at essential imprinted genes in babies born following ART versus those conceived naturally. This meta-analysis included 18 studies and reported that the frequency of imprinting disorders in ART-born babies was 3.67 higher than in naturally conceived children. The authors concluded that a raised risk of imprinting alterations occurs in babies born following IVF and ICSI; nevertheless, there was limited evidence linking epigenetic alterations at imprinted genes and ART [63]. Another review describing results from eight studies on BWS and ART reported a significant positive relation among IVF/ICSI procedures and BWS with an increased relative risk of about 5.2 times [64]. However, the authors did not observe an association for either AS or PWS with IVF/ICSI, but rather a positive association with fertility problems. Regarding SRS, the number of children born following ART was small (*n* = 13); therefore, relevant significance for SRS incidences could not be inferred. A more recent epidemiological study investigated the risk of imprinting disorders in IVF babies conceived in Finland and Denmark, where the authors compared the incidence rate of PWS, SRS, BWS, and AS in IVF-conceived babies in Denmark (*n* = 45,393, born 1994–2014) and Finland (*n* = 29,244, born 1990–2014). They observed an increased odds for BWS (OR 3.07, 95% CI: 1.49–6.31) in ART-conceived children; however, no significant difference was evident for PWS, SRS, and AS [65]. Similarly, a nationwide study in Japan found a 4.46-fold increase in BWS and an 8.91-fold increase in SRS following ART, including several with aberrant DNA methylation at imprinted genes [66]. The effect of altered epigenetics marks on human health is just beginning to be elucidated. A notable shortcoming of most existing studies is that they are not properly controlled for paternal age, which could modulate the occurrence of epigenetic problems. In one study, Day and colleagues showed that sperm methylation patterns in older men differ from that of their own somatic cells and younger counterparts [67]. The changes in DNA methylation with aging could increase the risk of developing neurodevelopmental disorders in the resulting offspring [68]. Additionally, changes in the promoter regions of genes containing CpG islands in sperm from older men might alter the function of genes associated with schizophrenia, bipolar disorders, and autism, thus increasing the risk of these disorders in children of elderly fathers [69,70]. Further research will help clarify whether ART-induced epigenetic changes will affect future offspring’s growth, development, and health. The following sections will discuss specific procedures applied during MAR treatments to explain how particular treatments may lead to epigenetic dysregulation.

## 4. Spermatogenesis, Epigenetics, and Infertility

Male fertility depends on the production of healthy sperm cells by the testis. This process is known as spermatogenesis and can be described by three main steps: first, the mitosis with the multiplication of the spermatogonia, then meiosis to reduce the number of chromosomes from diploid to haploid, and finally the spermiogenesis, which indicates the successful maturation of round spermatids into spermatozoa [71]. All of these processes are linked together and are responsible for normal sperm production; any alteration during spermatogenesis may cause a reduction in sperm quantity and quality. Recent evidence indicates that the dynamic of epigenetic reprogramming and their regulatory systems are fundamental for normal spermatogenesis. Any disturbances of these epigenetic regulations might result in different infertility stages, which could be transferred to future generations [25,72]. Abnormal DNA methylation is linked with changes in histone formations, dysregulation of lncRNA, and abnormal protamination, which might induce male infertility. Along these lines, histone modifications have been investigated in mature sperm. Ben Maamar and coworkers examined the alterations in DNA methylation during the early stage of gametogenesis from primordial germ cells (PGCs) to sperm. Several DNA methylation regions at the different developmental stages were analyzed. The study recognized a compelling cascade of epigenetic changes during the early developmental stages, indicating alterations to regulate gene function and expression during gametogenesis [73]. Furthermore, even after spermatogenesis is completed with the formation of the sperm cells, extra maturation takes place in the epididymis [74,75]. The sperm cell, following the release into the seminiferous tubules and the rete testes, will cross the efferent ducts into the epididymis, where further maturation occurs. During this passage, the epididymal cells produce specific proteins acquired by the sperm to achieve motility after ejaculation. Therefore, the sperm’s capability to achieve motility is mainly gained during epididymal transit [76,77]. Epigenetic regulation during epididymal maturation of the sperm cells remains to be clarified. Although the sperm nuclei are transcriptionally inactive due to the DNA compaction associated with protamines, it has been reported that environmental chemicals such as DDT or vinclozolin might induce epigenetic alterations, especially DNA methylation between caput and cauda epididymal sperm stage [78,79]. Indeed, during sperm epididymal maturation, histone modification and DNA methylation took place as additional epigenetic regulation, critically important for the sperm’s function and formation [80].

## 5. ICSI for Male Factor Infertility

Standard IVF has been successfully used since the birth of Louise Brown in 1978. However, its results are suboptimal, and the risk of total failed fertilization (TFF) is considerably high with abnormal or poor-quality sperm samples [12]. Thus, injection of a single sperm into an oocyte cytoplasm, which is capable of fertilization and can develop into a healthy baby, has become the most applicable fertilization method for couples with severe male factor infertility and is often applied for a variety of non-male factor infertility. Palermo and co-authors performed the ICSI technique for the first time following an accident during subzonal insemination (SUZI). The novel technique emerged and was quickly introduced worldwide for male factor infertility, without rigid validation [12,13]. The group published the first sets of injections performed on oocytes collected from four women. They obtained 31 fertilized oocytes and 15 embryos. After embryo transfer, four pregnancies to full term were described [12]. A couple of years later, the ICSI technique was applied with sperm aspirated from the epididymis in azoospermic patients. Tournaye and colleagues reported the first successful series in 12 patients: they described a fertilization rate of 58%, and five pregnancies were obtained out of ten fresh embryo transfers [81]. In the same year, 1994, the first report was published showing the efficacy of ICSI using sperm collected surgically from the testis [82]. Testicular sperm extraction (TESE) was introduced for patients with obstructive azoospermia. 

### 5.1. Oligoasthenoteratozoospermia

Although ICSI should be encouraged mainly in severe male infertility, it can be challenging to establish when a male factor is compulsory for the ICSI technique. Standard semen assessment is performed to confirm the severity of male infertility and advise ICSI, but it is well reported that sperm analysis has limitations; for example, it does not assess the function and physiology of the sperm, and genetic or epigenetic assessment [83]. Sperm number, morphology, and motility are typically evaluated to decide on the ICSI procedure rather than standard IVF insemination [84]. It is worth mentioning that high-quality studies investigating pregnancy outcomes and live birth rate (LBR) between ICSI and IVF in couples with oligoasthenoteratozoospermia are still missing. However, a study published in 2005 by Shuai and collaborators explored these concerns. The authors observed no differences between the two insemination procedures (IVF and ICSI) in fertilization, implantation, and pregnancy rates in couples undergoing ART with men diagnosed with moderate oligoasthenoteratozoospermia [85]. Sperm morphology is another parameter broadly used to choose for ICSI. In 1986, Kruger and colleagues suggested using strict criteria for sperm abnormalities and advising ICSI when the proportion of normal sperm in the ejaculate was <4% [86]. Additional studies confirmed this evidence and proposed that at least 5% of sperm is needed to be morphologically normal to obtain an acceptable fertilization rate using standard IVF [87,88]. Therefore, ICSI rather than IVF has been routinely recommended in patients with reduced sperm morphology (<5%) [89]. Despite that, a study published by Hotaling and co-workers reported that performing ICSI or IVF has no difference in pregnancy outcomes in patients with severe teratozoospermia. This study evaluated around 3000 IVF/ICSI treatments. Results indicated that the odds of clinical pregnancy in couples in which the male partner had severe teratozoospermia did not differ regardless of whether conventional IVF or ICSI was performed [90]. Another key sperm parameter is motility, and ICSI has been strongly suggested when there are few motile sperm or absolutely no sperm motility in the ejaculate (asthenozoospermia). In this case, it is essential to execute a sperm viability test, as injection of uncharacterized immotile sperm could reduce fertilization and embryo development rates [91]. Various laboratories have reported the use of pentoxifylline or theophylline to increase the selection of viable sperm to increase ICSI outcomes [92].

### 5.2. Azoospermia

The term azoospermia indicates the absence of sperm cells in the ejaculate. It affects around 1% of the general male population and about 15% of infertile men [93]. There are two different types of azoospermia: obstructive and non-obstructive. In obstructive azoospermia, normal and complete spermatogenesis is typically found, and sperm can be surgically collected from the testis [94]. By contrast, non-obstructive azoospermia is associated with the testicular alterations that result in the failure of sperm production. Typical testicular histopathological features in males with non-obstructive azoospermia include germ cell aplasia, maturation arrest, or hypospermatogenesis. The procedures mostly applied to collect sperm from azoospermic patients are percutaneous acquisition and open surgery [95]. Following sperm retrieval, ICSI can be applied to achieve oocyte fertilization [96].

### 5.3. Antisperm Antibodies

The presence of seminal antisperm antibodies (ASAs) is typically associated with a gap or rupture of the blood–testis barrier in the reproductive tract, which can be linked with several conditions [97]. However, elevated levels of ASAs in semen samples are observed in about 5–12% of men undergoing ART, and might negatively affect fertility, reducing sperm motility, capacitation, acrosome reaction, and oocyte sperm bounding [98]. A meta-analysis published by Zini and co-authors, including more than 4000 cycles, examined the relation between ASA and pregnancy outcomes in couples undergoing ART. Results showed that among men with high levels of ASAs, performing standard IVF insemination or ICSI was equally efficient in terms of pregnancy rates [97].

### 5.4. ICSI and Sperm DNA Fragmentation (SDF)

DNA fragmentation test is applied to assess the breakage of DNA strands inside the sperm head. This diagnostic test can predict fertility and normal embryo development and pregnancy outcomes than routine semen analysis parameters [99,100]. With the use of probes, sperm DNA breaks can be deeply scrutinized and quantified with the aid of fluorescence/optical microscopy or flow cytometry [100]. Sperm DNA fragmentation (SDF) is generally induced by oxidative stress resulting from environmental and lifestyle factors such as smoking, genital tract infections, obesity, and nutrition [101]. Moreover, SDF is frequently detected in men with infertility issues (e.g., varicocele), and it is more prevalent in those individuals than in fertile counterparts [102,103]. Scientific evidence indicates that a high level of SDF impairs the probabilities of success following ART [104,105]. Therefore, the analysis of SDF may be applied when deciding between ICSI and IVF as the method of fertilization. A study published by Simon and colleagues in 2017, including about eight thousand cycles, found that clinical pregnancy rates were higher in patients undergoing IVF whose male partners had low SDF levels [106]. Therefore, it seems reasonable to suggest ICSI rather than standard IVF in couples with high SDF undergoing ART treatments. However, a concern to consider is the risk of miscarriage, which appears to increase among couples undergoing ART who report sperm with high SDF, irrespective of IVF or ICSI. In a review including 11 trials and 1549 IVF and ICSI treatments, the authors showed the odds of miscarriage raised by 2.48-fold among men with high SDF [107]. Other studies have confirmed these findings showing that SDF was responsible for high pregnancy loss regardless of the methods applied for fertilization (ICSI or IVF) [108,109]. Altogether, these data support the notion that ICSI is superior to IVF as a method of fertilization in couples undergoing ART with high SDF; however, the risk of pregnancy loss is a concern with ICSI using ejaculated sperm [104,110].

### 5.5. Globozoospermia

This condition is described by the entire lack of the acrosomal vesicle in the sperm head, with alteration of the nuclear membrane, and midpiece defects, resulting in a round-shaped sperm head. It is an uncommon condition involving a small percentage of infertile men (about 0.1%) [111]. Despite having normal sperm count and motility, globozoospermic sperm cannot fertilize the oocyte: therefore, ICSI remains the favorable option available. However, a systematic review published by Rubino and colleagues showed low fertilization and pregnancy rates adopting ICSI due to the reduced capability of the sperm to activate the oocyte cell and induce zygote formation and embryo development [112].

## 6. Use of ICSI for Couples with Partners Having Semen Analysis within Reference Ranges

One of the first Cochrane review papers was published in 2004 by van Rumste and collaborators to investigate whether ICSI improves LBR compared to IVF in couples whose male partners had semen analysis within reference ranges. The authors showed a significantly higher fertilization rate in the IVF group but no difference in pregnancy, miscarriage, or LBR than ICSI insemination [113]. Subsequently, Bhattacharya and co-workers performed a multicenter randomized controlled study comparing clinical outcomes after ICSI or traditional IVF in couples with male partners having semen assessment within references ranges. The study randomly assigned 415 couples and was performed in four UK IVF units. Their results showed that the fertilization rate was higher with IVF than with ICSI (58% versus 47%; *p* = 0.0001). Standard IVF insemination provided an implantation rate of 30% compared to 22% for ICSI (*p* = 0.03). No significant difference was observed regarding the clinical pregnancy rate between IVF and ICSI (33% and 26%, respectively). Moreover, the overall laboratory time used was significantly shorter with IVF than with ICSI (22.9 min versus 38.1) [114]. Dang and co-workers reported similar results. They randomized 1064 patients undergoing ART to ICSI technique (*n* = 532) or standard IVF insemination (*n* = 532). After the first embryo transfer, LBR was 35% in the ICSI group versus 31% for couples assigned to conventional IVF (*p* = 0.27). They found higher TFF with IVF (6%) than with ICSI (5%). The study concluded that in couples undergoing ART with a male partner having so-called normal semen parameters, ICSI did not increase LBR compared with conventional IVF [115].

### 6.1. Unexplained Infertility

Johnson and collaborators, in 2013, published a meta-analysis including about 12,000 sibling oocytes, demonstrating that ICSI is associated with higher fertilization rates (67.5%) compared to standard IVF (47.8%) in couples with unexplained infertility (*p* < 0.001) [116]. They also found a significantly higher TFF with IVF than with ICSI (*p* < 0.001). Another trial released by Bungum and colleagues included about 250 patients with unexplained infertility who had already performed three or more unsuccessful intrauterine insemination (IUI). The authors found a TFF in 25% of IVF cycles compared with 4% in ICSI cycles, and they advised dividing the oocyte between IVF and ICSI for couples with unexplained infertility [117]. In a large retrospective study of about 112,000 conventional IVF and 205,000 ICSI cycles, including patients with unexplained infertility, the embryo transfer cancellation rate was similar in both groups (ICSI: 8.0% and IVF: 8.2%). In addition, following fresh embryo transfer both implantation and LBR rates were lower in the ICSI group than in the IVF (23% versus 25.2%; *p* < 0.001, and LBR 36.5% versus 39.2%; *p* < 0.001) [118]. Similar results were observed in another investigation published by Foong and colleagues [119]. Collectively, there seems to be enough evidence to suggest ICSI reduces TFF rates and increases fertilization rates in couples with unexplained infertility undergoing ART. However, retrospective data from RCTs that randomly allocated patients with unexplained infertility to IVF or ICSI reported that the ICSI technique does not increase pregnancy outcomes and LBR [117,118,119].

### 6.2. Advanced Maternal Age

In a retrospective study accounting for almost 750 couples with women aged >40 years, without obvious male infertility, Tannus and collaborators showed an equivalent LBR between ICSI and IVF following fresh embryo transfer. In their study, however, more embryos were available for cryopreservation in the IVF group than in the ICSI counterpart (26.4% versus 19.7%, *p* = 0.04) [120]. A trial by Haas and collaborators also investigated the same matter, i.e., the role of ICSI in couples undergoing ART cycles with females aged ≥39 and males with sperm parameters within reference ranges. Sixty-nine couples were assessed in the trial, and the result showed the fertilization rate (72.4% versus 65.1%; *p* = 0.38), the average number of cleavage stage embryos (2.8 versus 2.4; *p* = 0.29), and the average top-quality embryos (1.7 versus 1.6; *p* = 0.94) were similar among the IVF and ICSI groups. The author stated that ICSI does not increase pregnancy outcomes in patients who experienced ART with advanced maternal age and so-called normal sperm parameters [121]. As ICSI is a less natural and more intrusive technique, which necessitates extra time, this limited positive effect favoring ICSI regarding TFF may not be enough to support ICSI as the first-line treatment for couples without male factor infertility. This evidence is in agreement with other reports recently published [120,121,122,123]. Table 2 depicts the broad ICSI application in male and non-male factor infertility.

## 7. Contemporary Use (and Overuse) of ICSI

Since its first use almost 30 years ago, the application of ICSI as a fertilization method has raised steadily, even though the percentage of infertile couples with severe male factors has not increased [125]. Thus, it seems evident that currently, ICSI is applied broadly, even though there is no clear evidence of its benefit in couples without male factor infertility [113,114,123,125,126]. Boulet and collaborators analyzed data on ART between 1996 and 2012 and reported increased use of ICSI from 36.4% in 1996 to 76.2% in 2012, even though male-factor infertility remained unchanged at about 36% of cycles [118]. Another trial published by Dyer and colleagues analyzing the worldwide data on ART performed between 2008 and 2010 found that ICSI was used as a fertilization method in about 67% of about 4.5 million cycles completed [123]. However, there is considerable variation according to countries; in Asia, ICSI is applied in about 55% of the treatments, 65% of cases in Europe, 85% of patients in Latin America, and almost 100% of patients in the Middle East [123]. Moreover, in a large retrospective study performed in Australia between 2002 and 2013, analyzing about 585 thousand ART cycles, the authors did not report any improvement when ICSI was used rather than standard IVF insemination for couples without obvious male infertility. They observed an LBR of about 10% lower with ICSI than IVF [122]. On this basis, one should ask why is ICSI preferred to standard IVF in routine practice for cases without a clear male factor? Possible factors to justify the broad ICSI application related to a general notion that ICSI reduces the risk of TFF. Naturally, fertilization failure is problematic to any couple undergoing ART; especially when counseling is not available and the physician is the person involved in delivering this bad news. In addition, in private settings, where the couple needs to pay for the treatment, the failed fertilization also represents a remarkable burden for the couple who will need to bear the costs of another cycle. The debate is ongoing [115] and the Practice Committee of the American Society for Reproductive Medicine (ASRM) has recently produced a committee opinion paper recommending against the extensive use of ICSI in couples undergoing MAR cycles without confirmed male factor infertility [124].

## 8. Fertilization Process (IVF/ICSI) and Risk of Epigenetic Modification

There are two methods used for oocyte fertilization: the standard IVF insemination where sperm and oocyte are placed together into the same culture dish, and the sperm will fertilize the oocyte on its own, and the ICSI technique, where an embryologist utilizes an inverted microscope and a micromanipulator with a narrow glass injection pipette collects, immobilizes a single sperm before slowly releasing it into the oocyte cytoplasm (Figure 3). As already discussed, ICSI was adopted in clinical practice without prior experimental testing or clinical validation in animal models. In vivo, the process of fertilization takes place in the oviduct. It follows physiological events, including natural sperm selection, capacitation, acrosome reaction, and membrane fusion before the sperm nucleus is released into the oocyte cytoplasm. Indeed, with ICSI, all these processes are surpassed [127]. Some evidence has suggested that ART techniques might alter epigenetic reprogramming and eventually embryo development [43,128]. As mentioned earlier, epigenetic control of gene expression plays a crucial role in sperm function and fertilizing ability. Adequate regulation of epigenetic mechanisms, such as DNA methylation, chromatin remodeling, histone modifications, and non-coding RNAs during the development of gonadal and spermatogenesis, is essential for normal sperm production and function. Epigenetic regulation could be modulated by external and internal factors, including environmental exposure, nutrition, and stress. On this basis, male infertility conditions affecting sperm quality have been suggested to influence the sperm epigenome. Indeed, aberrant DNA methylation at imprinted genes has been associated with abnormal spermatogenesis and male factor infertility. DNA hypomethylation at the *H19* gene locus in sperm has been linked to oligozoospermia and azoospermia [129,130]. Kobayashi and colleagues analyzed DNA acquired from about 100 infertile men and reported disruption in paternal methylation in 14.4% of patients and abnormal maternal imprint in 20.6% of patients. The majority of these defective samples were in men with oligospermia. The authors concluded that sperm from infertile patients might be more likely to transmit incorrect imprints to their offspring [131]. Marques and collaborators [132] analyzed 24 infertile men and observed an altered *H19* methylation profile, suggesting a correlation between aberrant genomic imprinting and hypospermatogenesis. The authors concluded that spermatozoa from oligozoospermic men might bring a higher risk of transmitting imprinting alterations. ICSI could overcome infertility in the above cases, but the risk of transferring the abnormal sperm genome and epigenome to the embryo is not eliminated. Embryos with disrupted methylation features might inherit these abnormalities paternally. In fact, some studies have reported DNA methylation defects in embryos generated by ART (Table 3). It has been hypothesized that the ART process itself could be the cause impacting the methylation processes, including, for example, the altered hormonal environment associated with OS, the use of epigenetically immature gametes for fertilization, and the embryo culture conditions. Despite the above observations, the evidence regarding an increased frequency of epigenetic diseases in ART children remains equivocal. A retrospective cohort study, published by Whitelaw and collaborators and analyzing children born between 2002 and 2008, measured the DNA methylation in paternally expressed gene 3 (*PEG3*), insulin-like growth factor 2 (*IGF2*), *SNRPN*, long interspersed nuclear element 1 (*LINE-1*), and the insulin gene (*INS*) and reported no significant differences in term of DNA methylation, compared with children naturally conceived [133]. Another study, performed by Rancourt and co-workers investigated the methylation levels of *GRB10*, *MEST*, *H19*, *SNRPN*, and *KCNQ1*, as well as *IGF2DMR0*, and found that ART has no disruption effects on those genes; therefore, the conclusions were reassuring for infertile couples undergoing MAR treatment [53]. Additional studies have reported no significant epigenetic differences when comparing ART-born babies with those naturally conceived [134,135]. Nevertheless, some studies have shown that the placenta is more susceptible to epigenetic alterations when compared to embryos and can therefore be used as an alternative to measuring early epigenetic alterations affecting the embryo [56,136,137,138]. For example, placentas from ICSI children, but not IVF, were reported to have global *H3K4me3* differences compared to those of naturally conceived children [139]. The debate is still ongoing, as some authors report no significant difference between epigenetic modification after IVF or ICSI and natural conception [14,46,140], whereas others consider that epigenetic dysregulation of specific imprinted genes may increase the risk of disorders in babies conceived following ART. Further clarifications are urgently required to determine whether epigenetic errors or environmental and lifestyle risk factors contributed by the father or mother may be associated with imprinting disorders.

### 8.1. In Vitro Culture Associated Risks

In vitro fertilization has been ordinarily applied for decades in couples with infertility issues and has been considered a safe practice. However, recent trials have reported an association between ART procedures and an increased risk for adverse perinatal outcomes. Animal models provided evidence suggesting that imprinting in oocytes and embryos is sensitive to environmental changes. Several reports have shown the effects of in vitro culture on gene expression in preimplantation embryos in different mammals [151,152,153]. Epigenetic marks are acquired during the first phase of gametogenesis, the formation, and migration of PGCs into the gonadal ridge [154], and subsequently during the first days of embryo development. Correct establishment of epigenetic patterns is essential for embryo development. Indeed, the typical morphological assessment of gametes and embryos’ quality during ART cannot identify epigenetic alterations [155]. Several studies have shown disrupted methylation at several imprinted genes due to in vitro culture in certain media [156,157,158,159,160]. A comprehensive study by Schwarzer and co-authors analyzed IVF procedures compared to in vivo controls. In total, 5735 fertilized mouse oocytes were in vitro cultured or into the female oviduct and assessed for developmental parameters at the blastocyst stage. The authors reported that culture media might promote modifications in cellular, developmental, and metabolic pathways [161]. Similar results were found by Gad and colleagues investigating the effect of different culture media on the transcriptome profile of bovine preimplantation embryo until the blastocyst stage [162]. A few additional studies have explored the effects of culture media in human preimplantation embryos. Kleijkers and co-workers cultured human embryos in two different types of media typically used in ART. They observed differential expression of 951 genes involved in apoptosis, metabolism, protein processing, and cell cycle regulation [163]. A more recent study found differentially expressed genes following in vitro culture of human embryos; however, expression differences were higher due to maternal age and developmental stage. The authors concluded that they could not confirm whether the observed differences between embryos cultured in different media are caused by factors that were not examined and that further research is needed to validate those results [164]. Another example of the possible adverse effects of in vitro culture on embryo development is cattle with LOS following in vitro culture of ruminant embryos [62,165]. A study published by Chen and colleagues highlighted the concern that in vitro culture and ART induces misregulation of several imprinted genes in the kidney, brain, and liver of LOS fetuses, where the magnitude of overgrowth is associated with the number of epigenetically altered imprinted genes [166].

### 8.2. Oxygen Tension

In vitro culture is probably one of the most critical factors affecting epigenetic reprogramming, and oxygen concentration is the leading environmental factor affecting epigenetic alterations [167]. Since the 1950s, research has been conducted to determine oxygen concentrations in the female reproductive tract. For several decades, ART laboratories have been culturing embryos under oxygen concentration of around 20%. Later, it was established that oxygen tension in the female reproductive tract of mammalian species is between 2–8% [168], which indicates that embryos develop in vivo under low oxygen concentrations [169,170]. Thus, the IVF laboratory started to apply in vitro culture at low tension of 5%, similar to physiologic tensions in the female reproductive tract. Recently, the culture at an ultra-low oxygen concentration of 2–3% was postulated [171,172,173]. In the cytoplasm, oxidative stress resulting from the accumulation of reactive oxygen species (ROS) is likely to impair embryo development and implantation potential. In vitro culture of human embryos at reduced oxygen tension is an important feature to retain physiological evolution and increase reproductive competence. Placentas derived from in vitro culture at 20% oxygen concentration displayed a more significant difference in DNA methylation than those obtained from in vivo conceptions. In comparison, investigations on placentas obtained from in vitro 5% oxygen culture conditions did not show significant differences from those obtained from natural conceptions [129]. Several studies on mammals, including humans, suggested adverse effects of atmospheric oxygen levels on embryo development [172] and changes in the proteome, transcriptome, and epigenome [26,169]. Moreover, there is evidence supporting in vitro culture of human embryos at 5% levels, rather than 20%, to improve pregnancy outcomes [171,172,173,174]. A recent multicenter trial on 1563 oocytes confirmed that incorporating antioxidants in the culture media significantly increases embryo viability, implantation, and pregnancy rates, possibly via oxidative stress reduction [175]. Similarly, a Cochrane review meta-analysis stated that, compared with 20% oxygen concentration, embryos cultured at a low oxygen concentration of 5% yielded higher probabilities of IVF/ICSI success, ongoing clinical pregnancy, and live birth [176].

## 9. Conclusions

Since its introduction in 1992, ICSI has allowed many couples to overcome the burden of infertility. Nowadays, ICSI is widely used to remedy male and non-male factor subfertility. Despite its superiority over conventional IVF in couples with male factor infertility, its advantages over IVF among couples without a clear male factor are yet to be demonstrated. The overuse of ICSI should be carefully evaluated, given its potential genetic and epigenetic risks. Limited evidence suggests that babies born following the ICSI procedure have a raised risk of congenital malformations, chromosomal abnormalities, and altered reproductive hormonal profiles than naturally conceived children. Despite that, the link between ART, including ICSI, and epigenetic modifications increasing the risk of diseases in offspring, both in early and adult life, remains equivocal. Although some studies have suggested a possible link between ART and epigenetic defects, it is largely unknown whether these observations are associated with OS and luteal phase support regimens, ICSI as a method of fertilization, in vitro culture manipulations, or the cause of parental subfertility. Given the long-lasting effects on future generations’ health of early life conditions and epigenetic modifications, there is an urgent need for large-scale follow-up studies on the health of ART-born children not only at delivery but also at different time points into adulthood.

## Figures and Tables

**Figure 1 jcm-11-02135-f001:**
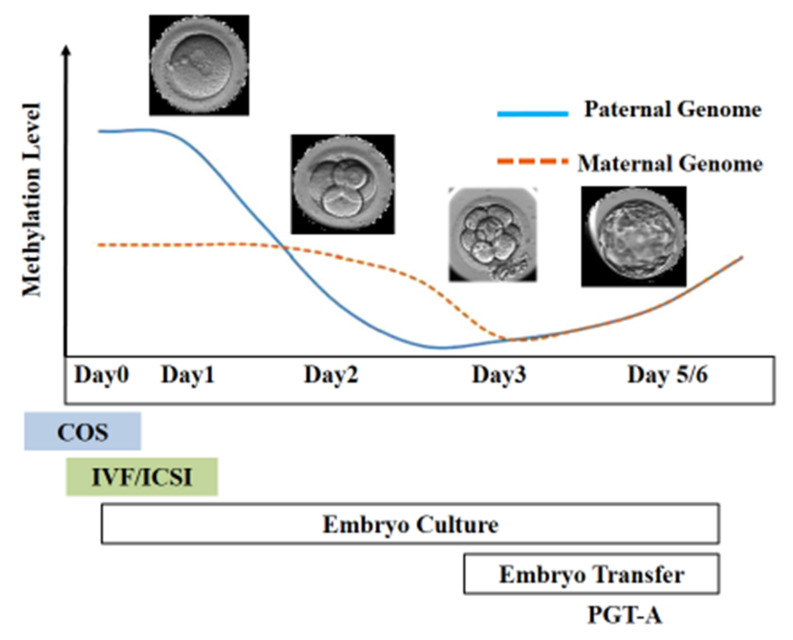
DNA methylation and epigenetic reprogramming during the early stage of embryo development. The paternal genome undergoes active demethylation post-fertilization, whereas the maternal genome is passively demethylated.

**Figure 2 jcm-11-02135-f002:**
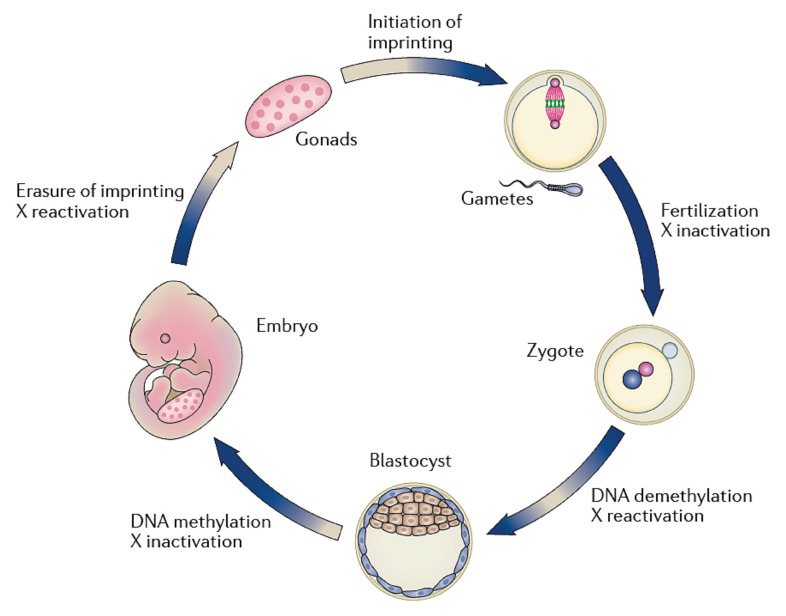
The epigenetic reprogramming cycle. The two major waves of epigenetic reprogramming occur during gametogenesis and after fertilization. During gametogenesis, most parental epigenetic marks are erased and re-established at the time of oogenesis and spermatogenesis. A second epigenetic reprogramming wave occurs soon after fertilization with a fast, active paternal demethylation and a slower, passive maternal demethylation. New methylation patterns are established at the blastocyst stage in the inner cell mass, while the trophectoderm stays relatively unmethylated. Adapted with permission from Ref [43].

**Figure 3 jcm-11-02135-f003:**
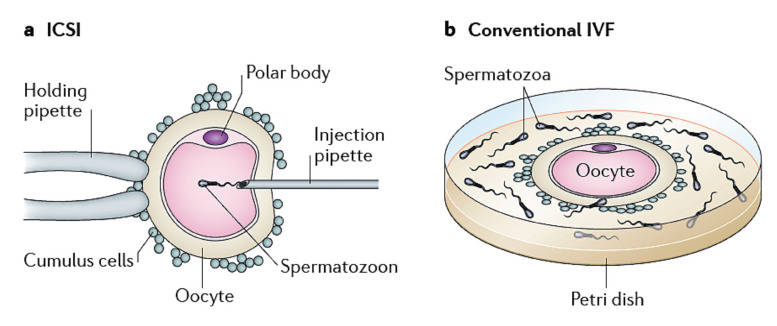
Assisted fertilization methods. (**a**) Intracytoplasmic sperm injection (ICSI) involves the injection of a single spermatozoon into an oocyte cytoplasm using a glass micropipette. (**b**) Standard in vitro fertilization (IVF), where oocytes are incubated with sperm in a Petri dish, and the sperm cell fertilizes the oocyte naturally. Reprint with permission from Ref [43].

**Table 1 jcm-11-02135-t001:** Studies comparing DNA methylation and imprinting/epigenetic diseases in ART infants and natural conceived infants. AGA, appropriate for gestational age; ART, assisted reproductive technology; DMRs, differentially methylated region; DNA, deoxyribonucleic acid; ICSI, intracytoplasmic sperm injection; IVF, conventional in vitro fertilization; MI, methylation indices; NC, naturally conceived; NR, not reported; OI, ovulation induction; RNA, ribonucleic acid; SD, standard deviation; SGA, small for gestational age. Adapted with permission from Ref [43].

Study	Design	Study Group (*n*)	Control Group (*n*)	Outcome Measures	Association	
					ART vs. NC Infants	ICSI vs. IVF Infants
Gomes et al., 2009 [44]	Prospective cohort	ART infants (18)	Negative controls: healthy NC infants (30). Positive controls: Beckwith-Wiedemann syndrome NC infants (3)	Gene studied: *KvDMR1* Genomic DNA was obtained from peripheral blood (12 of 18) and placenta (6 of 18) in ART infants; umbilical cord and placenta samples (8 of 30) and peripheral blood (22 of 30) in negative controls; peripheral blood samples (3 of 3) in positive controls	Hypomethylation at *KvDMR1* was observed in 3 of 18 clinically normal infants conceived by ART (2 conceived through IVF and 1 through ICSI). Nevertheless, discordant methylation was observed in three dizygotic ART twins. Normal methylation was observed in negative controls and hypomethylation was observed in positive controls.	Hypomethylation was found in both IVF and ICSI infants, suggesting that hypomethylation may not be restricted to a specific method of fertilization. Furthermore, infertility cause was not associated with hypomethylation, thus suggesting that hypomethylation may not be restricted to the presence of male factor infertility
Kanber et al., 2009 [45]	Prospective cohort	Small for gestational age ICSI infants (19)	Normal weight NC infants (29)	Genes studied: *KCNQ1OT1, PEG1, PEG3, GTL2, IGF2, H19, PLAG1* Genomic DNA was obtained from buccal smears	Hypermethylation of *KCNQ1OT1* and borderline hypermethylation of *PEG1* in one ICSI child only. ICSI was used due to male factor infertility (oligozoospermia), but the parents of the affected child had normal methylation patterns. The other studied ICSI children had normal methylation patterns	NR
Tierling, et al., 2010 [46]	Prospective cohort	ART infants (112; 35 IVF and 77 ICSI infants)	NC infants (73)	Genes studied: *KvDMR1, H19, SNRPN, MEST, GRB10, DLK1/ MEG3 IG-DMR, GNAS, NEsP55, GNAS NESPas, GNAS XL*-alpha-s, *GNAS* Exon1A Genomic DNA was obtained from maternal peripheral blood, umbilical cord and placenta	ART infants did not show a higher degree of imprinting variability. However, the mean methylation indices (MI) for one *DMR (MEST)* were higher in maternal peripheral blood (mean MI ± SD: 0.40 ± 0.03) and umbilical cord (0.41 ± 0.03) of IVF infants compared to NC infants (0.38 ± 0.04; *p* = 0.02, maternal peripheral blood and 0.38 ± 0.03; *p =* 0.003, umbilical cord).	The mean methylation indices for one *DMR (MEST)* were higher in maternal peripheral blood (mean MI ± SD: 0.40 ± 0.03) and umbilical cord (0.41 ± 0.03) of IVF infants than ICSI infants (0.37 ± 0.04; *p =* 0.0007, maternal peripheral blood and 0.38 ± 0.03; *p =* 0.003, umbilical cord)
King et al., 2010 [47]	Prospective cohort	ART infants (22)	NC infants (31)	Gene studied: *XCI* Genomic DNA was obtained from cord blood	Mildly skewed *X* chromosome inactivation *(XCI)* was present in 2 of 22 (9.1%) ART infants and 2 of 31 (6.5%) control infants. Extremely skewed *X* chromosome inactivation was present in 2 of 22 (9.1%) ART infants and 0 of 31 control infants. Neither difference was statistically significant; however, there was a trend toward a higher mean percentage of skewed *X* chromosome inactivation among ART infants	No risk difference for *XCI* skewing between ICSI (13) and IVF (9) groups (75.7% vs. 65.4%)
Turan, et al., 2010 [48]	Prospective cohort	ART infants (45)	NC infants (56)	Genes studied: *IGF2/H19, IGF2R* Genomic DNA was obtained from cord blood, cord and placenta	Aberrant methylation patterns at the *IGF2/H19* locus were more common in ART infants.	NR
Wong, et al., 2011 [49]	Prospective cohort	ART infants (77; 25 IVF-AGA, 7 IVF-SGA, 32 ICSI-AGA and 13 ICSI-SGA)	NC infants (12; 7 NC-AGA and 5 NC-SGA)	Genes studied: *H19, IGF2* Genomic DNA was obtained from placenta in all cases and umbilical cord in 7 of 77 ART infants	No significant differences in mean methylation between ART infants and NC infants. Mean ± SD methylation values were 44.68% ± 4.18% in NC-AGA and 44.63% ± 3.60% in NC-SGA.	No significant differences in mean methylation between IVF vs. ICSI infants. Mean ± SD methylation values were 45.52% ± 4.86% in IVF-AGA, 47.25% ± 5.77% in IVF-SGA, 45.64% ± 6.06% in ICSI-AGA, and 42.73% ± 4.39% in ICSI-SGA.
Li et al., 2011 [50]	Prospective cohort	ART twins (29)	NC twins (30)	Genes studied: *H19, IGF2, PEG1, KvDMR1* Genomic DNA was obtained from umbilical cord in all cases	Methylation indices were not significantly different between ART twins (45.68%) and NC twins (42.88%) in paternally methylated *H19/IGF2 DMRs*, nor were these indices different between ART twins (51.14%) and NC twins (50.67%) in maternally methylated *KvDMR1/PEG1 DMRs.*	NR
Feng et al., 2011 [51]	Prospective cohort	ART infants (60; 30 IVF and 30 ICSI)	NC infants (60)	Genes studied: *L3MBTL, PEG10, PHLDA2, PWCR1, SNRPN, UBE3A, TP73, GNAS, MEG3* Genomic DNA was obtained from umbilical cord blood and peripheral blood	The expression levels of *PEG10* (*p =* 0.018) and *L3MBTL* (*p =* 0.000) were significantly higher in ART infants than NC infants. The levels of PHLDA2 (*p =* 0.000) in ART infants were significantly lower than NC infants	NR
Oliver et al., 2012 [52]	Prospective cohort	ART infants (66; 34 IVF and 32 ICSI)	NC infants (69)	Genes studied: *H19, SNRPN, KCNQ1OT1, IGF2* Genomic DNA was obtained from peripheral blood samples	No significant differences in percentage of methylation between ART and control groups	NR
Rancourt et al., 2012 [53]	Prospective cohort	ART infants (59) and infants conceived by OI (27)	NC infants (61)	Genes studied: *GRB10, MEST, SNRPN, KCNQ1, H19, IGF2* Genomic DNA was obtained from umbilical cord blood and placenta tissue in all cases	Significant differences in median methylation levels were observed comparing infants conceived by OI to NC infants: in the placenta for H19 (40.2% OI vs. 44.6% NC; *p <* 0.0001), in the cord blood for *KCNQ1* (43.6% OI vs. 42.3% NC; *p =* 0.003), and in both cord blood (42.5% OI vs. 40.4% NC; *p* = 0.047) and placenta (43.2% OI vs. 41.1% NC; *p =* 0.005) for *SNRPN.* Likewise, significant differences in median methylation levels were observed comparing infants conceived by ART to NC: in the placenta for *H19* (43.4% ART vs. 44.7% NC; *p =* 0.01) and *SNRPN* (42.1% ART vs. 40.4% NC; *p =* 0.008), in the cord blood for *KCNQ1* (42.9% ART vs. 42.3% NC; *p =* 0.02). Additionally, *MEST* had lower methylation levels in the placenta of ART group than NC (48% vs. 51.4% *p <* 0.0001). Despite of that, differences in methylation levels did not translate into differences in overall gene expression	NR
Puumala, et al., 2012 [54]	Prospective cohort	ART infants (67)	NC infants (31)	Genes studied: *H19, KvDMR, IGF2, IGF2R* Genomic DNA was obtained from peripheral blood and buccal smears	No significant differences in the percentage of methylation between ART infants and NC infants.	NR
Hiura et al., 2012 [55]	Nationwide epidemiological study and prospective cohort study	ART infants diagnosed with imprinting diseases (6). One infant diagnosed with Beckwith-Wiedemann syndrome (ICSI) and five diagnosed with Silver-Russell syndrome (IVF)	NC infants diagnosed with imprinting diseases (16). Six infants diagnosed with Beckwith-Wiedemann syndrome and ten infants diagnosed with Silver-Russell syndrome.	Genes studied: *H19, KCNQ1OT1, PEG1, GRB10, INPP5F, ZNF597, FAM50B, ZDBF2, PEG10, ZNF331, NESPAS* Genomic DNA was obtained from blood or buccal smears	A 10-fold increase in the frequency of Beckwith-Wiedemann syndrome (0.03% vs. 8.6%) and Silver-Russell syndrome (0.02% vs. 9.5%) in ART infants compared to NC infants (based on the 2009 population of Japan: 127,510,000). No significant differences were found in the methylation ratios of infants diagnosed with imprinting diseases conceived by ART or naturally	NR
Nelissen et al., 2013 [56]	Prospective cohort	ART infants (35; 5 IVF and 30 ICSI)	NC infants (35)	Genes studied: *MEST, PEG3, KCNQ1OT1, SNRPN, H19, DLK1, MEG3* Genomic DNA was obtained from placenta	Hypomethylation at *H19* and *MEST* and increased RNA expression of *H19* were observed in placentas from ART infants but not in placentas of NC infants.	NR
Sakian et al., 2015 [57]	Prospective cohort	ART infants (107; 56 through IVF and 41 through ICSI)	NC infants (22)	Genes studied: *H19, IGF2* Genomic DNA was obtained from placenta	Both IVF and ICSI placental tissue displayed higher *H19* expression than controls (1.8 and 1.9 fold higher, respectively). *IGF2* was significantly decreased in both IVF and ICSI groups (0.8 and 0.7 fold lower, respectively) when compared with the NC group	No differences were observed between IVF and ICSI placentas
Melamed et al., 2015 [58]	Prospective cohort	IVF infants (10)	NC infants (8)	Genome-wide approach in cord blood (total 27,578 CpG sites)	A total of 733 (2.7%) of the CpG sites were differentially methylated between the 2 groups (*p* < 0.05), with an overall significant higher relative hypomethylation in IVF infants than NC infants (*p* < 0.001)	NR
Vincent et al., 2016 [59]	Case-control study	ART infants (182; 101 IVF and 81 ICSI)	NC infants (82)	Genes studied: *PLAGL1, KCNQ1OT1, PEG10, CDKN1C, IGF2, YWHAZ, KvDMR1, LINE-1* Genomic DNA was obtained from umbilical cord blood and/or placenta	DNA methylation at *PLAGL1* was significantly higher in IVF (47.9%) than ICSI (45.9%) and NC (45.9%) cord blood (*p <* 0.01). *PLAGL1* expression was lower in both IVF (*p <* 0.01) and ICSI (*p =* 0.02) cord blood groups than in NC infants.	DNA methylation at *PLAGL1* was significantly higher in IVF (47.9%) than ICSI (45.9%) (*p <* 0.01). No differences were found in DNA methylation between IVF and ICSI for *KvDMR1* and *LINE-1* in cord blood and placenta as well as *PLAGL1* and *PEG10* in placenta villi

**Table 2 jcm-11-02135-t002:** ICSI or IVF as fertilization methods applied in male and non-male factors infertility. ICSI: intracytoplasmic sperm injection; IVF: in vitro fertilization; OAT: oligoasthenoteratozoospermia. Adapted with permission from Ref [43].

Infertility Factor	Method of Fertilization	Study [Ref]
**Male factor infertility**		
Azoospermia	ICSI mandatory	[86,92,94,95]
Moderate OAT	IVF and ICSI equally effective	[83,85]
Severe OAT	ICSI highly recommended	[81,91,92]
Absolute asthenozoospermia	ICSI mandatory	[91,92]
Antisperm antibodies	IVF and ICSI equally effective	[106,107,108]
Sperm DNA fragmentation	ICSI recommended	[106,107,110]
Globozoospermia	ICSI mandatory	[111,112]
**Non-male factor infertility**		
General non-male factor	Equally effective, slightly in favor of IVF	[113,114,115]
Preimplantational genetic testing	ICSI highly recommended	[84,124]
Unexplained infertility	Equally effective	[117,118,119]
Poor responders	Equally effective, slightly in favor of IVF	[119,120]
Poor oocyte quality	Equally effective, slightly in favor of IVF	[117,121,122]
Advanced maternal age	Equally effective, slightly in favor of IVF	[120,122,123]

**Table 3 jcm-11-02135-t003:** Studies that examined epigenetic modifications in infertile males. Adapted with permission from Ref [43].

Study	Type	Study Group (*n*)	Control Group (*n*)	Outcome Measures	Association
Hartmann et al., 2006 [141]	Pilot study	Men diagnosed with spermatogenic arrest at the level of spermatogonia (3) and spermatocytes (6)	None	*H19* genomic DNA was obtained from different germ cell types derived from seminiferous tubules exhibiting impaired spermatogenesis	No abnormal *H19* methylation in spermatogonia or spermatocytes in azoospermic men
Peng et al., 2018 [142]	Pilot experiment	Oligoasthenozoospermic men (OA:39) Asthenoteratozoospermic men (AT:36)	Normozoospermic men (50)	Aberrant methylation of the imprinted genes *H19* and *SNRPN* (small nuclear ribonucleoprotein polypeptide *n*)	The mean methylation level of *H19-ICR* in the OA group (78.66%) and the AT group (84.56%) was significantly lower than in the *n* group (88.51%, *p* < 0.001) Similarly, the mean methylation level of *SNRPN-ICR* in the OA group (8.36%) and the AT group (10.37%) was significantly higher than in the *n* group (6.32%, *p* < 0.001)
Kobayashi et al., 2007 [131]	Pilot experiment	Infertile couples with oligozoospermic men (18)	Infertile couples with normozoospermic men (79)	Genes studied: *H19, GTL2, PEG1, LIT1, ZAC, PEG3, SNRPN* Genomic DNA was obtained from sperm	Abnormal paternal methylation (*H19* and *GTL2*) imprint in 14 patients and abnormal maternal methylation (*PEG1, LIT1, ZAC, PEG3*, and *SNRPN*) in 20 patients. The occurrence of abnormal methylation at the *H19* and *GTL2* was significantly increased in oligozoospermic patients when compared with normozoospermic patients
Marques et al., 2008 [143]	Cohort	Oligozoospermic men (20)	Normozoospermic men (5)	Genes studied: *MEST, H19* Genomic DNA was obtained from sperm	Infertile males with a sperm count below 10 × 10^6^/mL displayed defective methylation of imprinted genes (*H19* hypomethylation and *MEST* hypermethylation)
He et al., 2020 [144]	Cohort	Asthenospermic men (16) Oligozoospermic men (3) Oligoasthenospermic men (11)	Normozoospermic men (8)	Differentially methylated regions *(DMRs)* of imprinted genes: *H19, GNAS, MEG8*, and *SNRPN*	*DMRs* of imprinted genes *H19, GNAS, MEG8*, and *SNRPN*, were different in the abnormal semen groups. *MEG8 DMR* methylation in the asthenospermic group was significantly increased
Kobayashi et al., 2009 [145]	Cohort	Aborted samples from women subjected to ART treatment and parental sperm (78)	Aborted samples from non-ART women and parental sperm (38)	Genes studied: *H19, GTL2, PEG1, KCNQ1OT1, ZAC, PEG3, SNRPN*, and *XIST* Genomic DNA from trophoblastic villi of aborted samples and parental sperm	Seventeen of 78 ART aborted samples presented abnormal DNA methylation at one or more imprinted gene. In 7 of these cases, the same imprinting errors were present in the parental sperm
Marques et al., 2010 [132]	Cohort	Azoospermic men (24) 5 with anejaculation, 5 with secondary obstructive azoospermia, 5 with primary obstructive azoospermia, 9 with nonobstructive azoospermia due to hypospermatogenesis	None	Genes studied: *H19, MEST/PEG1* Genomic DNA was obtained from human testicular sperm	Methylation at *H19* and *IGF2* was significant reduced in nonobstructive azoospermic patients
Boissonnas et al., 2010 [146]	Cohort	Teratozoospermic men (19) Oligoasthenoteratozoo-spermic men (22)	Normozoospermic men (17)	Genes studied: *H19*, *IGF2* Genomic DNA was obtained from sperm	In the teratozoospermia group, 11 of 19 patients presented a loss of methylation at variable CpG positions either in the *IGF2 DMR2* or in both the *IGF2 DMR2* and the 6th *CTCF* of the *H19 DMR*. In the oligoasthenoteratozoospermia group, 16 of 22 patients presented a severe loss of methylation of the 6th *CTCF*, which was associated with sperm concentration
Kobayashi et al., 2017 [147]	Cohort	Moderate oligozoospermic men (40) Severe oligozoospermic men (30)	Normozoospermic men (151)	DNA methylation patterns of 3 paternally and 19 maternally methylated *DMRs*	Aberrant methylation levels in 25 of the 151 patients (16.6%) with normozoospermia, 9 of the 40 patients (22.5%) with moderate oligozoospermia and 21 of the 30 patients (70.0%) with severe oligozoospermia
Song et al., 2021 [148]	Cohort	80 cases showing impaired sperm DNA integrity	Normozoospermic men (86)	Methylation status of 257 CpG sites among *H19* and *SNRPN* and four non-imprinted genes related to male infertility (*MTHFR**,* *GSTM1**,* *DAZL**,* and *CREM**)*	Differential methylation found in 43 CpG sites of 6 genes: *H19*, *SNRPN*, *MTHFR*, *DAZL*, *GSTM1* and *CREM* The imprinting genes were associated with relatively higher rates of differentially methylated CpG sites (28.21% in *H19* and 41.38% in *SNRPN*) than the non-imprinting genes
Khambata et al., 2021 [149]	Case-control study	Sperm collected from male partner of 112 couples with history of recurrent pregnancy loss (RPL)	Normozoospermic prover fertile men (106)	DNA methylation status of selected imprinted genes such as *IGF2-H19 DMR, IG-DMR, MEST, ZAC, KvDMR, PEG3, PEG10*, and *SNRPN*	In the RPL group, a significant decrease in the global sperm 5mC levels and significant decrease in DNA methylation at three CpG sites in *LINE1* promoter was found For *IGF2-H19 DMR* and *IG-DMR*, a significant decrease in sperm DNA methylation at specific CpG sites was observed in RPL group
Tang et al., 2018 [150]	Cohort	135 men with idiopathic male infertility, including normozoospermia (*n* = 39), moderate oligozoospermia (*n* = 45), and severe oligozoospermia (*n* = 51)	Fertile control Normozoospermic men (59)	DNA methylation status of CpG sites within the differentially methylated regions *(DMRs)* of three imprinted genes, *H19, GNAS*, and *DIRAS3*	Aberrant methylation patterns of imprinted genes were more prevalent in idiopathic infertile males, especially in patients with oligozoospermia Infertile males with aberrant methylation patterns of imprinted genes showed a lower global methylation levels, which was not statistical significance (*p* = 0.13)

## Data Availability

Not applicable.

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
