# Peer review of "Contemporary Use of ICSI and Epigenetic Risks to Future Generations"

_jcm, 2022, doi:10.3390/jcm11082135_

Round 1

Reviewer 1 Report

The review on Contemporary use of ICSI and epigenetic risks to future generations, by Romualdo Sciorio, Sandro C. Esteves is highly interesting and of crucial important for the ART and also for general medical and broader scientific community.

I would like to offer some critical points how to improve this manuscript.

1) Please target the abstract and specify the goal of this review and findings that are discussed in the manuscript. The abstract could be better organised, to give justice to the topic. I suggest to modify it severely and give clear indications what will be discussed in the text of the review.

2) background is well written, but the references should be extended, especially in epigenetic modifications, that are exclusively delivered by male pronucleus and contribute to the early embryogenesis. There should be introduced more the importance of male epigenome, therefore the importance of selecting the best sperm for ICSI, which is yet rather absent and limited by lack of advance methods for sperm selection.

These three papers are recommended to be discussed with the others:

doi: 10.1186/s13148-015-0058-4

doi: 10.4161/epi.21556

doi: 10.1007/s10815-018-1354-7

3) the figure, showing the importance of epigenetic code delivered by sperm, and alter utilised by oocyte and reprogrammed after fertilization should be included, as that is the main topic for the manuscript. The Figure 1 and 2 are too general in my opinion, and they are not very illustrative and novel.

4) I do not think that one chapter much be just on ICSI historical facts, this chapter can be merged with background and that would be probably desired, as the background should cover all parts of the text.

5) chapter 8 deserves more discussion and not only list of references and percentage. If the authors say overuse, which I agree, they should provide opinion, and also discussion on the current data collected and say why.

6) chapter 9 should be more scientifically approached, with implementation of publications on that topic as recommended above and others. It is crucial to show evidence, that sperm pathologies carry also aberrant epigenome, which is overcome by ISCI.

7) table 2 is not complete, must be updated and thoroughly revised.

8) Conclusion deserves update in same style like an future updated abstract. There should be provided concrete summary of the article, directly to the point based on the evidence. This style conclusion is not strong.

Author Response

Thanks very much for your time and for the review performed on our paper. 

Thanks for the comments that you provied, which help us to improve the paper. You have the file with our replay. Please have a look at the file attached.

Best Regards

Reviewer 2 Report

Comments to the Authors:

This review evaluates the literature regarding the history of and current use of ICSI as well as the possible risks of epigenetic abnormalities that may affect perinatal and long term outcomes of children born via ICSI.  The authors begin by discussing the history of ART and the common epigenetic mechanisms.  The review further delves more deeply on the studies that have evaluated epigenetic changes that may occur via ART.  Lastly the authors discuss the indications for ICSI over IVF and the risks of epigenetic changes.  The manuscript is overall well written, and the authors provide a detailed review of literature.   

Please consider the following comments when revising your manuscript:  

General Comments:

  • Good review!

Comments:

Background

  • For references 5-8 it is important to mention whether or not the cited studies looking at ART vs naturally conceived children controlled for parental age as for example there is a strong association of paternal age and autism spectrum disorders in natural conception with advance parental age
  • Lines 57-60 read a bit clunky

Imprinting alteration following ART

  • In this section again it is important to note whether the studies controlled for paternal age or not.

Contemporary use (and overused) of ICSI

  • Change Overused to Overuse

Conclusion

  • Line 540-541 doesn’t read well.

Author Response

Thank you very much for your time to rwview our paper, and thanks for the comments yor provided which help to improve the mansucript.

Please found or replay in the file attached. 
